# COMBI-r: A Prospective, Non-Interventional Study of Dabrafenib Plus Trametinib in Unselected Patients with Unresectable or Metastatic BRAF V600-Mutant Melanoma

**DOI:** 10.3390/cancers15184436

**Published:** 2023-09-06

**Authors:** Carola Berking, Elisabeth Livingstone, Dirk Debus, Carmen Loquai, Michael Weichenthal, Ulrike Leiter, Felix Kiecker, Peter Mohr, Thomas K. Eigentler, Janina Remy, Katharina Schober, Markus V. Heppt, Imke von Wasielewski, Dirk Schadendorf, Ralf Gutzmer

**Affiliations:** 1Department of Dermatology, Uniklinikum Erlangen, Comprehensive Cancer Center Erlangen—European Metropolitan Region Nürnberg, Friedrich-Alexander University (FAU), 91054 Erlangen, Germany; markus.heppt@uk-erlangen.de; 2Department of Dermatology, University Hospital Essen, and German Cancer Consortium (DKTK), Partner Site Essen, 45147 Essen, Germany; elisabeth.livingstone@uk-essen.de (E.L.); dirk.schadendorf@uk-essen.de (D.S.); 3Department of Dermatology, Nuremberg General Hospital—Paracelsus Medical University, 90419 Nuremberg, Germany; debus@klinikum-nuernberg.de; 4Department of Dermatology, Klinikum Bremen-Ost, Gesundheit Nord gGmbH, 28205 Bremen, Germany; carmen.loquai@gesundheitnord.de; 5Department of Dermatology, Skin Cancer Center, University Hospital Schleswig-Holstein, Campus Kiel, 24105 Kiel, Germany; mweichenthal@dermatology.uni-kiel.de; 6Department of Dermatology, University Hospital Tuebingen, 72076 Tuebingen, Germany; ulrike.leiter@med.uni-tuebingen.de; 7Department of Dermatology and Venereology, Vivantes Klinikum Berlin Neukölln, 12351 Berlin, Germany; felix.kiecker@vivantes.de; 8Department of Dermatology, Elbe Kliniken Buxtehude, 21614 Buxtehude, Germany; peter.mohr@elbekliniken.de; 9Department of Dermatology and Allergy, Skin Cancer Center Charité, Charité—Universitätsmedizin Berlin, 10117 Berlin, Germany; thomas.eigentler@charite.de; 10Novartis Pharma GmbH, 90429 Nuremberg, Germany; janina.remy@novartis.com (J.R.); katharina.schober@novartis.com (K.S.); 11Department of Dermatology, Skin Cancer Center Hannover, Hannover Medical School, 30625 Hannover, Germany; vonwasielewski.imke@mh-hannover.de; 12Comprehensive Cancer Center (Westdeutsches Tumorzentrum), University Hospital Essen, Essen & National Center for Tumor Diseases (NCT); NCT-West, Campus Essen & Research Alliance Ruhr, Research Center One Health, University Duisburg-Essen, 45147 Essen, Germany; 13Department of Dermatology, Johannes Wesling Medical Center, Ruhr University Bochum, 44801 Minden, Germany; ralf.gutzmer@rub.de

**Keywords:** melanoma, BRAF mutation, MAPK pathway, dabrafenib, trametinib, brain metastases, tumor dynamics

## Abstract

**Simple Summary:**

The outcome of patients with advanced melanoma has profoundly improved over the last 15 years. Novel medications blocking BRAF, a protein involved in stimulating cell division, or inhibiting immune checkpoints associated with T-cell activation significantly improved overall survival. Mutations of the BRAF protein kinase cause cells to make an abnormal protein that promotes tumor growth. In patients with BRAF V600 mutations, dabrafenib and trametinib blocking BRAF V600 and MEK, respectively, have demonstrated improved efficacy in two large clinical trials (COMBI-d, COMBI-v) compared to blocking BRAF signaling with single agents. This led to the approval of dabrafenib plus trametinib for the treatment of patients with advanced melanoma. The study (COMBI-r) presented here investigated the use of dabrafenib plus trametinib in everyday clinical practice. COMBI-r confirms the data from COMBI-d and COMBI-v and provides additional data in patients with brain metastases or previous treatments who had been excluded from the pivotal trials.

**Abstract:**

Combined BRAF/MEK-inhibition constitutes a relevant treatment option for *BRAF*-mutated advanced melanoma. The prospective, non-interventional COMBI-r study assessed the effectiveness and tolerability of the BRAF-inhibitor dabrafenib combined with the MEK-inhibitor trametinib in patients with advanced melanoma under routine clinical conditions. Progression-free survival (PFS) was the primary objective, and secondary objectives included overall survival (OS), disease control rate, duration of therapy, and the frequency and severity of adverse events. This study enrolled 472 patients at 55 German sites. The median PFS was 8.3 months (95%CI 7.1–9.3) and the median OS was 18.3 months (14.9–21.3), both tending to be longer in pre-treated patients. In the 147 patients with CNS metastases, PFS was similar in those requiring corticosteroids (probably representing symptomatic patients, 5.6 months (3.9–7.2)) compared with those not requiring corticosteroids (5.9 months (4.8–6.9)); however, OS was shorter in patients with brain metastases who received corticosteroids (7.8 (6.3–11.6)) compared to those who did not (11.9 months (9.6–19.5)). The integrated subjective assessment of tumor growth dynamics proved helpful to predict outcome: investigators’ upfront categorization correlated well with time-to-event outcomes. Taken together, COMBI-r mirrored PFS outcomes from other prospective, observational studies and confirmed efficacy and safety findings from the pivotal phase III COMBI-d/-v and COMBI-mb trials.

## 1. Introduction

Advanced melanoma is an aggressive, life-threatening disease arising from the oncogenic transformation of melanocytes, currently causing 57,000 deaths per year worldwide [1]. Compared to former Globocan estimates [2,3], mortality rates are beginning to decrease slightly [4]. However, at a global level, incidence rates of melanoma continue to rise with population growth, population aging and increased UV exposure [5,6].

Therapeutic advances in pharmacotherapy may have contributed to the decline in mortality rates, along with ongoing skin cancer screening efforts and behavioral changes in developed countries, such as reduced exposure to and better protection from UV radiation. 

Since 2011, immune checkpoint inhibitors (ICIs) and novel agents targeting the mitogen-activated protein kinase (MAPK) pathway have changed the dismal outcomes for advanced melanoma patients by prolonging overall survival (OS) considerably [7]. Particularly, the combination of ICI, namely the PD-1 inhibitor nivolumab plus the CTLA-4-inhibitor ipilimumab on the one hand, and of BRAF inhibitors plus MEK inhibitors on the other, set unprecedented benchmarks of efficacy in patients with unresectable or metastatic melanoma. 

In the phase III study Checkmate-067, nivolumab and ipilimumab demonstrated an OS rate of 52% for treatment-naïve patients at 5 years, and the median OS was 72.1 months (95% confidence interval (CI) 38.1—not reached (NR)) [8,9]. For the combination of the BRAF inhibitor dabrafenib and the MEK inhibitor trametinib, the pooled analysis of the pivotal phase III trials, COMBI-d and COMBI-v, showed a 5-year OS rate of 34% (95% CI 30–38) and a median OS of 25.9 months (22.6–31.5) in previously untreated advanced melanoma patients with a BRAF V600E or V600K mutation [10]. For the combination of the BRAF inhibitor vemurafenib and the MEK inhibitor cobimetinib, a 5-year OS rate of 31% (25–37) [11] and for the combination of encorafenib and binimetinib, a 5-year OS rate of 35% were reported [12].

Since clinical trials for registrational purposes are usually carried out in selected patient populations, they do not provide comprehensive evidence for a broad patient population. Patients with unfavorable prognostic factors or risk factors like active brain metastases requiring in part the use of corticosteroids, which may hamper patients’ follow-up in a clinical trial, are usually excluded from pivotal trials. Some insight in such difficult-to-treat melanoma patients was gained through the COMBI-mb trial, recruiting within four cohorts 125 patients in total with either asymptomatic (pre-/untreated), rare mutation types (other than BRAF V600) or symptomatic brain mutations arising from melanoma [13].

Thus, results from registrational trials cannot be easily extrapolated to patient populations in community settings treated under routine clinical practice [14,15]. Hence, the generation of real-world evidence (RWE) from large and unselected patient populations is highly relevant to inform on patient population characteristics and outcomes, and to complement phase III trial results [16]. Here, we report effectiveness, safety and quality-of-life data from COMBI-r, a large, multi-center, prospective, single-cohort, non-interventional study conducted to generate real-world data from skin cancer centers in Germany. 

## 2. Materials and Methods

### 2.1. Study Design and Population

COMBI-r is a prospectively planned observational cohort study, recruiting adult patients with unresectable or metastatic BRAF-V600-positive melanoma. Patients were enrolled independent of therapy line and treated according to the Summary of Product Characteristics (SmPC). Patients with brain metastases could be enrolled as well as patients with hepatic or renal impairment or other comorbidities. Prior treatment with any MEK inhibitor monotherapy or with any BRAF-MEK combination therapy was not allowed. Patients could not participate in this study in case of treatment initiation with dabrafenib plus trametinib more than 12 weeks before giving informed consent. Enrolment in other clinical trials was not allowed during the treatment period.

### 2.2. Study Objectives

The main objective of COMBI-r was to collect real-world effectiveness, quality-of-life and tolerability data from a large, diverse and unselected patient population that received dabrafenib and trametinib per regular prescription in Germany. Therefore, patients with asymptomatic and symptomatic brain metastases as well as patients with an ECOG performance status (ECOG-PS) of ≥2 were enrolled among others. The primary objective to assess effectiveness was progression-free survival (PFS), and secondary objectives included disease control rate (DCR), best overall response (BOR) and OS. As further secondary objectives, changes in quality of life (QoL) under therapy were assessed, as well as the duration of therapy, reasons for and duration of dose modifications, and the frequency and severity of adverse events (AEs). 

### 2.3. Study Procedures and Assessments

The treatment visits followed the routine care scheme at each participating site or center, with documentation of effectiveness, safety and quality-of-life data taking place approximately every 3 months during the first year of therapy and then every 6 months until a patient no longer benefitted from therapy or developed unacceptable toxicity. Visits during the follow-up period occurred approximately 3 and 6 months after end of treatment, followed by further optional 3-month intervals. The duration of this study was estimated based on a 24-month recruitment plus approximately 24-month treatment period, i.e., until 80% of patients had either progressed or stopped the therapy, with a further 6 months of follow-up time to collect further PFS and OS events. The line of therapy served as a stratification factor for analyses. As per protocol amendment, the recruitment period was prolonged for one additional year in order to reach 500 patients by the end of this study.

Assessments of effectiveness were based on routine radiological and clinical evaluations. PFS was defined as time between start of dabrafenib plus trametinib therapy and either the date of first progression/relapse or until the date of death due to any reason. OS was calculated from start of dabrafenib plus trametinib combination therapy. Assessments of tolerability of therapy encompassed the documentation of all AEs, serious adverse events (SAEs), and the frequency and duration of dose reductions.

To assess dose intake patterns, drug administration data from patients’ charts, as primary data source, were converted into the electronic case report forms. To assess quality of life, patients were asked to complete on paper the FACIT-Fatigue scales (version 4) [17] and the EuroQol Five Dimensions (EQ-5D) questionnaire [18] from visit 1 to 9. As a supplementary tool for correlation analysis of study outcomes, investigators were asked to subjectively assess each patient’s tumor growth dynamics at baseline (visit 1) and end of treatment (visit 7, EoT). For this, the investigator categorized the patients’ state of disease at baseline into slow, intermediate and fast-growing tumor dynamics considering clinical stage, metastasis pattern, therapy line and lactate dehydrogenase (LDH) status. 

### 2.4. Statistical Analysis

Quantitative and qualitative data were analyzed descriptively. Kaplan–Meier estimate analysis was used for the assessment of time-to-event endpoints. All analyses were performed for the whole population as well as stratified by therapy line. Categorical variables are reported as frequency and percentage, whereas continuous variables are reported as means, standard deviation, median and range values. And 95% CIs were calculated for each category as applicable. 

### 2.5. Ethics Committee Approvals and Trial Registration

This study was approved by the local ethics committee of all participating sites; the German Federal Institute for Drugs and Devices (BfArM) as the competent regulatory authority was notified a priori about the conduct of this study. This non-randomized phase IV study was conducted in accordance with the applicable regulations in Germany for non-interventional studies; the decision to treat a patient with dabrafenib and trametinib had to be made independently by the investigator prior to enrolment into this study. Each subject had to provide written informed consent prior to enrolment. This study was registered (on 7 December 2016) at the German Clinical Trial Register, a primary register acknowledged by the World Health Organization registry network (DRKS-ID: DRKS00011387). The reporting of COMBI-r follows the STROBE recommendations [19].

## 3. Results

### 3.1. Patients 

From 12 October 2015 to 28 December 2018, 504 patients were enrolled at 55 German skin cancer centers. Thirty-two patients were excluded from the analysis due to deviations from the inclusion/exclusion criteria (Appendix A). As the most common protocol violation reason, 17 patients had been treated previously with a BRAF-MEK inhibitor combination other than dabrafenib plus trametinib or with an MEK inhibitor, including trametinib. Several of the 32 patients violated more than one inclusion/exclusion criterion; other reasons for protocol violation were lacking documentation of dabrafenib plus trametinib dose (n = 14) or treatment start more than 12 weeks prior to baseline, i.e., prior to start of documentation (n = 8), leaving in total 472 melanoma patients in the analysis population. A total of 450 patients (95.3%) of the analysis population completed visit 7 (end-of-treatment). 

At data cutoff (28 July 2021), 88 patients without progression or death were censored by date of their last visit (Appendix A). Patient characteristics at baseline are outlined in Table 1. Most patients (n = 300, 63.6%) received dabrafenib plus trametinib as first-line therapy; 172 (36.4%) patients were treated in later therapy lines. Patients receiving first-line treatment were on average 4 years older than those receiving later lines of therapy; male patients represented 55.5% of the study population. An amount of 49.6% of patients showed elevated LDH levels, and 91.8% of patients with documented ECOG-PS (73.7% of all patients) had an ECOG-PS of 0 or 1. Two-thirds of patients (63.1%) had a disease stage IV M1c; categorization followed the seventh edition of the American Joint Committee on Cancer (AJCC) staging criteria [20].

More than 80% of the second-line patients have received prior ICI therapy: most frequently PD-1 antibodies without additional CTLA-4 antibody therapy (48.5%), followed by combined PD-1 plus CTLA-4 therapy (30.6%). Detailed data of prior treatments are provided in Appendix A. During the treatment phase, 80 patients (16.9%) received concomitant radiotherapy, 16 patients (3.4%) surgical therapy and 5 patients (1.1%) radiosurgery. About one third of the patients received concomitant pharmacotherapy (first line 99 (33.0%); 59 (32.3%) in later lines), most often systemic corticosteroids and analgesics. Detailed data for concomitant therapies are provided in Appendix A. For concomitant drug as well as for non-drug therapy, multiple entries per patient were allowed in the case report form. 

### 3.2. Treatment Exposure

Patients received a starting dose of 150 mg of dabrafenib twice daily plus 2 mg of trametinib once daily as recommended by the protocol and the SmPC. However, in 50 patients (10.6%), dabrafenib 75 mg twice daily was the recorded starting dose. For dabrafenib and trametinib, the median exposition to therapy—including dose interruptions—was shorter in patients with first-line therapy (210 days) than in patients with later therapy lines (266 days). End-of-combination therapy was documented in 259 patients with first-line therapy (86.3%) and 153 patients (89.0%) with later therapy lines. The most frequent reasons for EoT were disease progression (first line 41.4%, later line 36.6%) and adverse event occurrence (first line 18.8%, later line 23.5%).

Mean daily doses of dabrafenib (first line: 293.1 mg, other line: 284.9 mg) and of trametinib (first line 1.9 mg, later line 1.9 mg) were similar in patients with the first and later lines of therapy. The dose of either dabrafenib or trametinib was reduced at least once in 116 patients (38.7%) in the first line and 84 patients (84.4%) in later lines of therapy. The median duration of dose reduction was similar for both strata (15.3 and 15.0 days). AEs were the most frequent reason for dose reductions (first line: 47.2%, later lines: 31.4%); however, reasons for dose reduction were not documented for 41.5% and 47.3% of patients, respectively.

### 3.3. Clinical Outcomes

The median follow-up time for the overall population, i.e., the time from treatment initiation to the last documented visit or date of censoring, was 13.5 months (range 0.0–59.6). At the cutoff date, 384 PFS events were documented (81.4%; 81.3% in first line, 81.4% in later lines), of which 321 were documented with disease progression and 63 were documented as death without a prior progression event.

#### 3.3.1. Effectiveness (Per Line of Therapy)

The median PFS was 7.7 months (95% CI: 6.6–9.1), 8.8 months (6.9–11.3) and 11.2 months (6.6–19.3) in patients in first-line, second-line or other-line therapies, respectively. Figure 1 (Panel A) shows the Kaplan–Meier curves of PFS stratified by therapy line. The overall median PFS in the analysis population was 8.3 months (7.1–9.3).

During the study period, 265 patients died (170 (56.7%) in first line, 95 (55.2%) in later lines). In most cases, death was related to progressive disease (first line 87.6%, other lines 94.7%). In five first-line patients (3.0%) and three patients (3.2%) in other therapy lines, death was not related to melanoma. There was no documented death due to treatment complications, adverse drug reactions or tumor-induced diseases.

Median OS was shorter in patients with first-line therapy (17.5 months (95% CI 13.3–21.3)) compared to patients with second-line (19.0 months (12.8–46.0)) or other-line therapies (20.6 months (13.7—NR)). Figure 1 (Panel B) shows the Kaplan–Meier curves of OS stratified by therapy line.

For 448 patients, BOR during therapy (assessed at EoT) per stratum was recorded (Appendix A). In the overall cohort, 108 patients (24.1%) had stable disease, 119 patients (26.6%) had a partial response and 54 patients (12.1%) had a complete response. Appendix A displays the radiological and clinical criteria underlying the routine clinical assessment of BOR. 

The DCR, indicative of an anti-tumor effect of dabrafenib plus trametinib therapy over a certain time (i.e., at 6, 12 or 24 months), was 79.9% at 6 months, based on disease status data of 402 patients. Twelve months after therapy start, 291 out of 315 patients (92.4%) have had at least one response or lasting disease stabilization reported (Appendix A).

#### 3.3.2. Effectiveness in Selected Subgroups

Subgroup analyses with respect to PFS and OS were performed for multiple subgroup variables. Appendix A show subgroup analyses for sex (slightly worse OS for male patients), for elevated LDH level (LDH increased: median OS 11.4 months (9.6–14.0); LDH not increased: ‘median NR’ (22.5—NR)), and for the presence of CNS metastases (present: median OS 10.9 months (95% CI 8.7–13.3); absent: 20.4 months (17.3–35.6)). Similarly, in patients with ≥3 organs affected by distant metastases, OS was significantly decreased (≥3 organs: median OS 10.8 months (9.1–12.5); 1–2 organs: 23.1 months (19.1–41.2)) (Appendix A). Due to the relevance and frequency of melanoma metastasis to the CNS, ad hoc subgroup analyses of PFS and OS by line of therapy were carried out (Figure 2, Panels A and B). In addition, the concomitant intake of corticosteroids by line of therapy (Appendix A) was analyzed. The PFS and OS medians of patients with active CNS metastases either requiring corticosteroids and of patients with CNS metastases not requiring corticosteroids are listed in Figure 2, Panel C. The respective Kaplan–Meier curves are displayed in the Appendix A.

We analyzed ad hoc the therapy-line-dependent time-to-CNS-metastasis (Figure 3). Medians were not yet reached, but both curves did not separate, indicating that the incidence of de novo brain metastases was the same regardless of treatment line.

#### 3.3.3. Investigator-Assessed Tumor Dynamics

Investigator-assessed tumor dynamics at visit 1 served as a helpful variable to identify patients at risk for a poor clinical outcome. For PFS as well as for OS, this subjective assessment correlated well with a poor (‘fast-growing tumor’) versus good (‘slow growing tumor’) outcome; the 95% confidence intervals of the time-to-event medians were not overlapping (Figure 4, Appendix A). 

#### 3.3.4. Therapy-Duration-Dependent Effectiveness

In line with the observed shorter exposure to therapy of patients receiving first-line therapy (see Section 3.2), the median time until end of treatment was 7.8 months (95% CI 6.9–9.2) shorter than in second-line patients (9.4 months (95% CI 7.4–12.0)) and in patients in later lines (10.3 months (95% CI 6.6–21.4)) (Appendix A). A progression-free survival of 10.3 months (95% CI 9.1–11.6) and overall survival of 22.6 months (95% CI 19.8–39.3) were reported in patients with a disease control under combination therapy of at least 6 months (Appendix A).

### 3.4. Tolerability

Adverse events of any cause occurred in 447 (94.7%) of the 472 patients analyzed. The distribution of AE and SAE showed a trend toward higher event rates in later lines of therapy (Table 2). In two-thirds of patients, investigators assumed AEs to be related to dabrafenib (N = 310, 65.7%) and/or to trametinib (N = 301, 63.8%). Similar to AE rates, treatment withdrawal rates related to either dabrafenib (162, 34.3% in total population) or trametinib (157, 33.3%) were higher in later lines of therapy, compared to first-line therapy (Table 2). Pyrexia of any grade, an adverse event of specific interest (AESI) associated and commonly observed with dabrafenib, was documented in 113 patients (23.9%). A decreased cardiac ejection fraction of any grade (10 patients, 2.1%) and increased lipase levels of any grade (13 patients, 2.8%) were the only other AESI reported in ≥2% of the overall safety population (N = 472).

Tumor progression was the most frequently reported SAE (N = 128 (27.1%) in total population). Since non-related AEs leading to death only had to be documented within 30 days after last treatment with dabrafenib plus trametinib, the numbers in Table 2 differ from the above-reported overall numbers of patients deceased during this study. No deaths were reported to be treatment-related or due to treatment complications. 

### 3.5. Quality-of-Life

At baseline, FACIT-F and EQ-5D utility index were completed by 358 (75.8%) and 385 (81.6%) patients, respectively. After Visit 2 (N_Valid_ = 224 (47.5%) and 234 (49.6%), respectively), completion rates began to drop. Twenty-four months after baseline, only 29 (6.1%) and 28 (5.9%) questionnaires, respectively, were completed. In comparison to visit 1, FACIT-F decreased on average by −2.06 in patients with first-line therapy and by −2.38 in patients with other therapy lines at the EoT period. The EQ-5D utility index did not indicate any changes in QoL over time. However, these findings should be interpreted cautiously because of the sharply decreasing number of patients with valid data for follow-up visits.

## 4. Discussion

Real-world data may narrow the evidence gap between clinical trials and clinical practice [14,15,21,22]. COMBI-r, a prospectively planned non-interventional clinical study, reports RWE from 472 melanoma patients enrolled at 55 German skin cancer centers. 

Few studies so far have reported prospectively collected RWE for combined BRAF/MEK inhibition. For reports on dabrafenib plus trametinib, we identified a French phase IIIB early access program (EAP) in 856 patients [23], a Dutch population study using data from a prospectively planned registry describing outcomes of 435 first-line patients [24], and a small Japanese post-marketing surveillance study [25]. For the combination of vemurafenib plus cobimetinib, only one completely published non-retrospective real-world study report from France was identified [26]. Apart from these prospective non-interventional studies, a number of retrospective studies and case series were published; however, due to the known constraints of retrospective studies, we refer in this discussion to the few published real-world and the previously cited confirmative trials on BRAF/MEK inhibition. To our knowledge, three large (comparable in size to COMBI-d/-v, i.e., >200 patients), retrospective observational case series were fully published so far [27,28,29].

### 4.1. Real-World Effectiveness of Dabrafenib Plus Trametinib

The median PFS in COMBI-r (8.3 months (95% CI 7.3–9.1)) was comparable to the French EAP (8.0 months (7.3–8.8)) in which 52.4% of patients were treatment-naive. For first-line use, the Dutch registry study reported a median PFS of 8.0 months [6.8–9.4], which is within the same range as first-line results from COMBI-r (median PFS, 7.7 months (6.6–9.1)). The pooled analysis of the registrational COMBI-d and COMBI-v trials—conducted in previously untreated patients—showed a median PFS of 11.1 months (9.5–12.8) [10]. For first-line vemurafenib plus cobimetinib, median PFS patterns among the French EAP (7.3 months (5.2–8.4)) and the registrational CoBRIM study (12.6 months (9.5–14.8)) indicated a ‘real-world’ effectiveness gap of similar magnitude [11,26]. These differences in PFS of around 3.5 to 4 months seem to represent the ‘real-world’ effect difference between selected trial populations and the clinical routine.

The median OS in COMBI-r for first-line use (17.5 months (95% CI 13.3–21.3)) was considerably lower than in COMBI-d and COMBI-v (25.9 months (22.6–31.5)) [11], but higher than in the Dutch registry study (11.7 months (10.3–13.5)) [24]. 

It is interesting to note in COMBI-r that for first-line patients, PFS and OS medians were shorter than for second-line patients, and even shorter than for patients receiving late-line dabrafenib and trametinib (Figure 1). This observation was consistent across the respective subgroup analyses (i.e., patients with CNS metastases, Figure 2). One factor explaining this somehow unexpected observation might be a selection bias, i.e., that patients selected for later therapy lines are the fittest ones. Second, a previous immune checkpoint blockade—more than 80% of the second-line patients have had prior ICI therapy—might have had a positive ‘priming’ effect on such targeted therapy. Third- and fourth-line patients in COMBI-r received on average even more than one prior ICI therapy. The hypothesis of the priming of *BRAF V600* mutant melanoma through prior immunotherapy was recently backed mechanistically by in vivo data [30]. Real-world experiences [31] and the randomized phase II SECOMBIT trial [32] and the phase III Dream-seq trial [33] also support such a hypothesis, although final results from large sequential use strategy trials are still awaited.

In COMBI-r, treatment exposure was lower and the duration of therapy was shorter in patients receiving dabrafenib plus trametinib as first-line therapy compared to later lines. Except for age (median 56.5 versus 52.2 years), there were no other baseline variables potentially explaining exposure differences among the 300 first- and 170 second- and other-line patients. Early changes in therapy due to multiple first-line alternatives or switching to another BRAF/MEK combination in case of intolerance may provide explanations for this observation. We assume, however, that the shorter treatment exposure in the first line affects the observed shorter PFS and OS medians for first-line dabrafenib plus trametinib. 

The BOR and DCR data of COMBI-r underscore the effectiveness of this BRAF/MEK combination in clinical routine use. The use of clinical response rates as an efficacy surrogate was recently backed by a meta-analysis showing that for targeted therapies, the response rates correlate strongly with PFS and well with OS [34]. A Dutch network-meta-analysis emphasized that dabrafenib plus trametinib is an effective and favorable therapy option in advanced melanoma, particularly with regard to the improvement in PFS [35].

### 4.2. Effectiveness in Brain Metastases

CNS metastases are common in melanoma and associated with a particularly poor prognosis, causing death in 60–70% of melanoma patients [36]. The high number of patients with CNS metastases in the COMBI-r cohort constitute, together with the CNS cohort of the French EAP (N = 275) and the phase II COMBI-mb trial population (N = 125), one of the largest BRAF/MEK-treated CNS metastasized cohorts analyzed so far [13,37]. 

Patients with CNS metastases in COMBI-r benefited from BRAF/MEK inhibitor therapy; the overall median PFS in COMBI-r (5.9 months (95% CI 4.8–6.4)) corresponds with the median reported in the French EAP (5.7 months (5.3–6.9)) [37]. In the COMBI-mb trial, patients with symptomatic brain metastases and no prior local brain therapy had a shorter median PFS (5.6 months (5.3–7.4)) compared to those with prior local therapy (7.2 months (4.7–14.6)) [13]. 

COMBI-r is one of the first RWE studies to report detailed PFS and OS outcomes for patients with brain metastases. In the 44 patients requiring systemic use of corticosteroids, the median PFS (5.6 months (95% CI 3.9–7.2)) was not affected (Figure 2, Panel C) compared to the 103 patients not requiring such concomitant therapy (5.9 months (95% CI 4.8–6.4)). However, the OS median for patients requiring corticosteroids was shorter, as one may expect (Figure 2, Panel C). Our data correspond with post hoc subgroup analyses of the COMBI-mb trial indicating that baseline corticosteroid use was independently associated with inferior clinical outcomes (intracranial response rate, PFS, OS) in patients with melanoma brain metastases [38].

Finally, a COMBI-r ad hoc analysis showed that, in patients without brain metastases at therapy start, the median time-to-CNS metastases was not reached during the 3-year observation period: neither for first- nor for later-line patients (Figure 3).

### 4.3. Investigator-Assessed Tumor Dynamics

Novel early markers of efficacy in oncology could help complement endpoints like ORR that summarize longitudinal tumor data into a single outcome measure (i.e., responder/non-responder) [39]. Similarly, novel patterns of tumor response to ICI also require a better understanding of tumor growth kinetics [40]. In our study, we prospectively requested clinicians to assess at baseline each patient’s tumor growth dynamics (slow, intermediate and fast) taking into account well-known prognostic markers for advanced melanoma [41,42]. The correlation of this investigator assessment with PFS and OS (Figure 4) suggests that such a simple, subjective assessment based on the physician’s clinical experience may predict patterns of progression, allowing up-front differentiation of patients with higher risk from those with lower risk of dismal outcomes. However, the explorative results from our study require further validation in prospectively planned clinical trials.

### 4.4. Safety Patterns

The safety profile of dabrafenib plus trametinib in COMBI-r was comparable to those reported in previous studies [10,43,44]. No new safety signals were observed and no treatment-related deaths were reported. The reporting of AEs, constituting a benchmark for data quality in a non-interventional study, was with 95% in a range quite near to COMBI-d and COMBI-v (98%) [10]. The AE-related therapy discontinuation rate of around one third of patients (Table 2) is higher than the 18% reported for the pooled long-term analysis of the Combi-d and -v trials [10]. Again, the ease of switching in clinical routine care to another BRAF/MEK combination or to other treatment options may explain the higher rate observed here. Pyrexia, as the most common AE associated with dabrafenib plus trametinib, however, was reported at a lower rate (23.9%) compared to the pooled analysis of COMBI-ad, -d and -v trials (61.3%) [45], most likely due to an improved pyrexia management. 

### 4.5. Strengths and Limitations of This Study

With 384 out of 472 patients having a documented tumor progression event at the cut-off date, COMBI-r data are mature and conclusive. The methodological variability in observational study designs and their varying legal and regulatory requirements render comparisons of data among non-interventional studies difficult [46]. For the present paper, we therefore limited data comparisons to the respective registrational (i.e., pivotal) studies on the one hand, and to prospectively conducted non-interventional studies on the other. 

However, as for clinical and observational studies in general, limitations must be acknowledged. Site selection always constitutes data bias [47]. In addition, investigators’ selection of patients who provide informed consent and participate in such a trial may result in a ‘convenience sample’, which might induce patient selection bias and subsequent interpretation bias. The assessment of clinical responses to therapy under real-world conditions constitutes another limitation: radiological examination is often performed outside the treating hospital. And even for in-house imaging, radiologists routinely do not provide their evaluations, by referring to RECIST criteria [48]. As a result, the assessment of progression in clinical practice builds upon radiological and upon clinical appraisals of progression. 

Similarly, the use of PFS, associated with RECIST-oriented computer tomography and/or magnetic resonance imaging, as a primary endpoint in a real-world effectiveness study is subjected to uncertainty. The schedules of PFS assessment may add further statistical uncertainty [49]. On the one hand, PFS allows for assessing outcomes of a given treatment independently from post-treatment therapies, which we could not comprehensively monitor, due to data privacy rules. But on the other hand, PFS is a good endpoint neither in single cohort studies nor in real-world studies [50,51,52]. Readers should therefore be aware that PFS may not be an appropriate primary endpoint in a non-interventional cohort study.

Finally, readers should take into account that treatment strategies have changed since COMBI-r enrolled its patients. Immunotherapy as a first-line therapy option is now recommended by clinical practice guidelines and many melanoma patients with cerebral metastases are treated today with nivolumab plus ipilimumab up-front [36]. This may further impact on the shown efficacy patterns of anti-BRAF and anti-MEK therapy and might result in an even stronger priming effect than with the immune checkpoint inhibitor monotherapy. These changes, as well as the availability of other BRAF/MEK combination regimens, should be considered in terms of the generalizability of the COMBI-r trial outcomes.

## 5. Conclusions

COMBI-r confirms that under real-world conditions, dabrafenib plus trametinib constitutes a safe and effective treatment for patients with advanced *BRAF-V600*-mutated melanoma. The combination is also effective in metastatic patients with one or multiple prior lines of therapy for *BRAF-V600*-mutated melanoma. In patients with brain metastases, meaningful PFS and OS medians were observed. COMBI-r assessed for the first time the risk and time patterns of de novo formation of new brain metastases under dabrafenib and trametinib, and medians were not reached. The clinician’s subjective up-front assessment of tumor dynamics allowed for reliably predicting PFS and OS outcomes of patients categorized at baseline to have a slow- or fast-growing tumor.

## Figures and Tables

**Figure 1 cancers-15-04436-f001:**
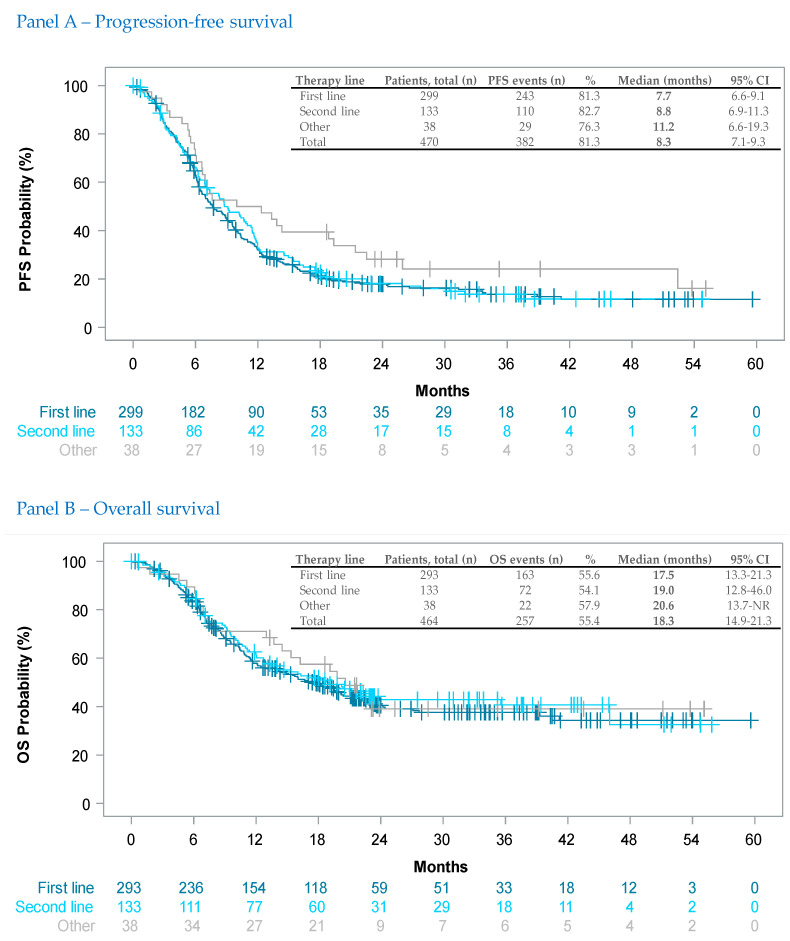
Kaplan–Meier estimates of progression-free survival Panel (**A**) and overall survival Panel (**B**) at 13.5 months follow-up. Vertical markers show censored patients. Cut-off date was 28 July 2021.

**Figure 2 cancers-15-04436-f002:**
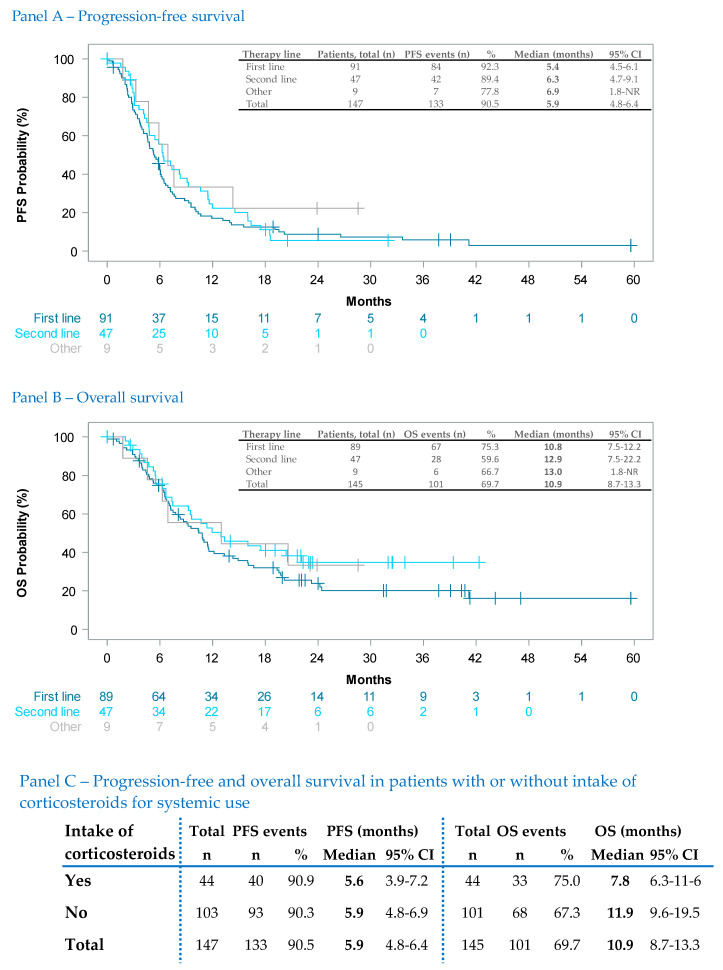
Ad hoc analysis of stage IV patients with CNS metastases according to line of therapy. Panel (**A**) and Panel (**B**) show Kaplan–Meier estimates of progression-free survival and overall survival, respectively. Vertical markers show censored patients. Panel (**C**) shows the tabulated PFS and OS medians for patients with CNS metastases requiring either systemic use of corticosteroids or not.

**Figure 3 cancers-15-04436-f003:**
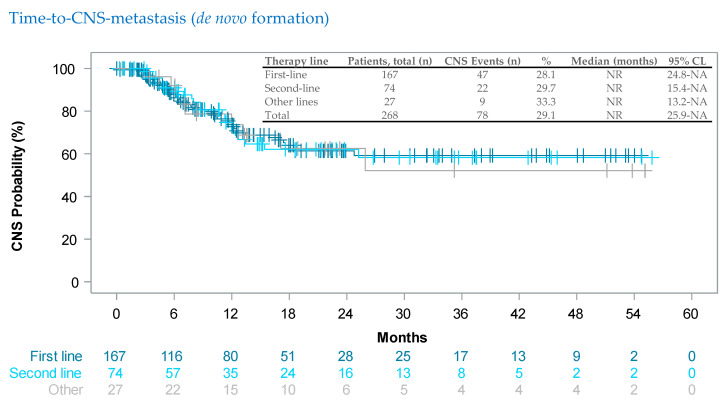
Kaplan–Meier estimates of the time-to-CNS-metastasis, i.e., the formation of de novo brain metastases after the start of the dabrafenib plus trametinib combination therapy. Vertical markers show censored patients. Cut-off date was 28 July 2021. Median follow-up was 13.5 months.

**Figure 4 cancers-15-04436-f004:**
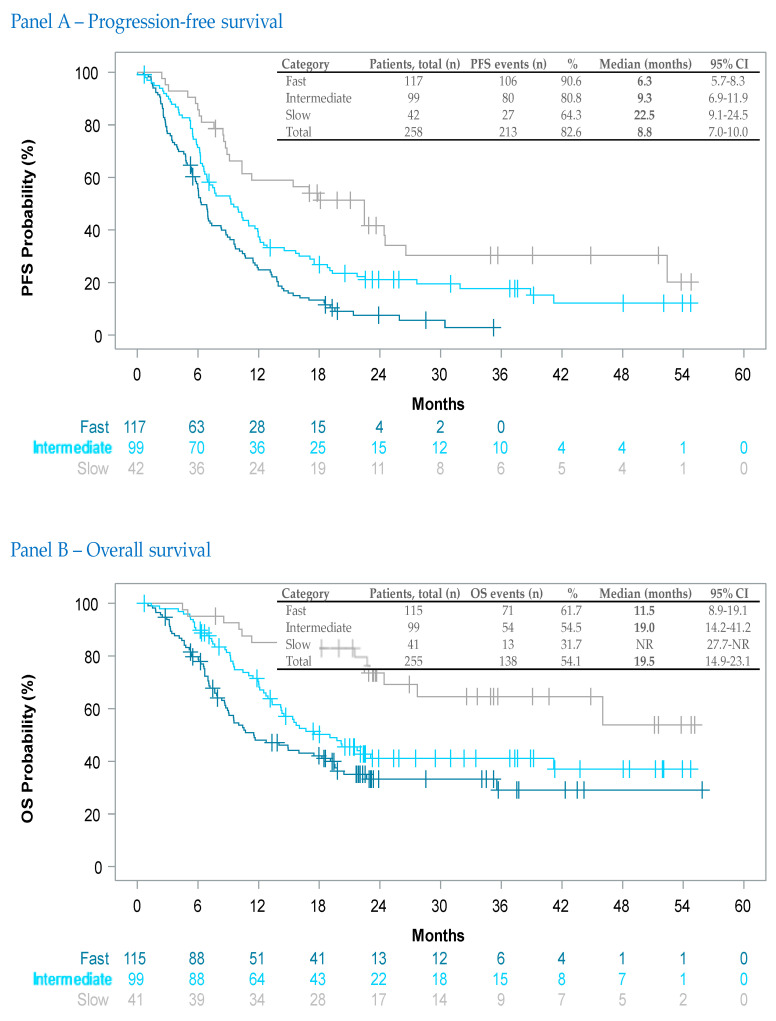
Kaplan–Meier estimates of progression-free survival Panel (**A**) and overall survival Panel (**B**) through investigator-assessed tumor dynamics. Vertical markers show censored patients. For this assessment, patients had to be categorized up-front (i.e., at visit 1) by the investigator considering patients’ clinical stage, metastasis patterns, therapy line and lactate dehydrogenase status.

**Table 1 cancers-15-04436-t001:** Baseline characteristics of patients (N = 472).

		First Line	Second Line	Other Lines	Total
		n (%)	n (%)	n (%)	n (%)
Total		300 (100.0)	134 (100.0)	38 (100.0)	472 (100.0)
Gender	Male	165 (55.0)	77 (57.5)	20 (52.6)	262 (55.5)
	Female	135 (45.0)	57 (42.5)	18 (47.4)	210 (44.5)
Age (at baseline)	Median (years)	61.0	56.5	59.7	59.2
	Range	24–89	21–87	28–86	21–89
	Missing data	0	0	0	0
ECOG PS	0	159 (53.0)	84 (62.7)	22 (57.9)	265 (56.1)
	1	57 (19.0)	17 (12.7)	9 (23.7)	83 (17.6)
	≥2	23 (7.7)	6 (4.5)	2 (5.2)	31 (6.6)
	Missing data	61 (20.3)	27 (20.1)	5 (13.2)	93 (19.7)
Clinical stage	IIIC	26 (8.7)	8 (6.0)	2 (5.3)	36 (7.6)
	IV	268 (89.3)	125 (93.3)	36 (94.7)	429 (90.9)
	Missing data *	6 (2.0)	1 (0.7)	0 (0.0)	7 (1.5)
LDH	Normal/decreased	102 (34.0)	44 (32.8)	17 (44.7)	163 (34.5)
	Elevated	152 (50.7)	69 (51.5)	13 (34.2)	234 (49.6)
	Missing data	46 (15.3)	21 (15.7)	8 (21.1)	75 (15.9)
Distant metastases	M0	29 (9.7)	9 (6.7)	2 (5.3)	40 (8.5)
	M1	10 (3.3)	4 (3.0)	0 (0.0)	14 (3.0)
	M1a	27 (9.0)	12 (9.0)	3 (7.9)	42 (8.9)
	M1b	35 (11.7)	26 (19.4)	8 (21.0)	69 (14.6)
	M1c **	192 (64.0)	81 (60.4)	25 (65.8)	298 (63.1)
	Mx	4 (1.3)	0 (0.0)	0 (0.0)	4 (0.8)
	Missing data	3 (1.0)	2 (1.5)	0 (0.0)	5 (1.1)
Affected organ systems ^†^	1	68 (26.3)	38 (31.4)	9 (25.0)	115 (27.6)
	2	69 (26.6)	26 (21.5)	15 (41.7)	110 (26.4)
	≥3	122 (47.1)	57 (47.1)	12 (33.3)	191 (45.9)
	Lung ^††^	156 (60.9)	76 (62.8)	21 (58.3)	253 (61.3)
	Bone ^††^	66 (25.6)	27 (22.3)	8 (22.2)	101 (24.3)
	Liver ^††^	94 (36.4)	37 (30.6)	12 (33.3)	143 (34.5)
	CNS ^††^	91 (35.3)	47 (38.8)	9 (25.0)	147 (35.4)
	Lymph node ^††^	115 (44.4)	54 (44.6)	15 (41.7)	184 (44.2)

Abbreviations: LDH, Lactate dehydrogenase; CNS, Central nervous system; MAb, Monoclonal antibody. * In five patients, last documented clinical stage was not up-to-date (i.e., with dates 6–30 months prior to start of COMBI-r, comprising stages I-IIIA); ** including two patients that had been classified as M1d according to novel AJCC 8 classification; ^†^ patients with clinical stage IV and documented distant metastases only (based on 416 patients, 6 missing data sets for metastases sites); ^††^ multiple entries possible.

**Table 2 cancers-15-04436-t002:** Overall safety and tolerability outcomes of dabrafenib-trametinib.

	First Line n (%)	Second Line n (%)	Other Lines n (%)	Total n (%)
Any adverse event	278 (92.7)	132 (98.5)	37 (97.4)	447 (94.7)
Any dabrafenib-related AE	186 (62.0)	96 (71.6)	28 (73.3)	310 (65.7)
Any trametinib-related AE	177 (59.0)	98 (73.1)	26 (68.4)	301 (63.8)
Dose interruption * (dabrafenib-related AE)	95 (31.7)	51 (38.1)	16 (42.1)	162 (34.3)
Dose interruption * (trametinib-related AE)	88 (29.3)	53 (39.6)	16 (42.1)	157 (33.3)
Any serious adverse event	190 (63.3)	104 (77.6)	25 (65.8)	319 (67.6)
Death from any cause	115 (38.3)	55 (41.0)	15 (39.5)	185 (39.2)

* Temporary and permanent (i.e., withdrawals) dose interruptions.

## Data Availability

Novartis is committed to sharing, with qualified external researchers, access to trial and patient-level data and supporting clinical documents from eligible studies. The final clinical study report (Version 8 April 2022, English language version) is available via the German Clinical Trial Register (DRKS): (accessed on 11 July 2023). Individual patient data (IPD) contain potentially personal information and are stored with the authors. Upon request, authors may share data in anonymized form with the editors or fellow scientists.

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
