# Peer review of "COMBI-r: A Prospective, Non-Interventional Study of Dabrafenib Plus Trametinib in Unselected Patients with Unresectable or Metastatic BRAF V600-Mutant Melanoma"

_cancers, 2023, doi:10.3390/cancers15184436_

Round 1
Reviewer 1 Report
Dear Authors,
Thank you for this interesting work.
Do you have any data related to the mutational landscape of these patients treated with anti-BRAF and anti-MEK: how many displayed mutations other than BRAF V600 E/K? Other BRAF mutations are rare but they do exist and constitute a terapeutic challenge. If you have some data on that point? And maybe data on the efficacy of targeted therapie in patients with rare mutations?
One limitation that could be discussed is the fact that the therapeutic strategies have changed since the enrolement period of 2015-2018. I feel that nowadays, immunotherapy is the prefereed first line and there has been an increasing use of IPI+NIVO. Besides, the majority of patients with cerebral metastases are now treated with IPI+NIVO as first line therapy. This may impact the efficacy of anti-BRAF+anti-MEK as a second line therapy because there might be a stronger priming effect than with the monotherapy.
Moreover there are now many prescription of enco+bini, the results may thus be different when integrating these new molecules.
Maybe these points could be discussed?
Best regards
Line 210: "have had"
Author Response
Please find our reply enclosed below as a word document

Reviewer 2 Report
The manuscript "COMBI-r: a Prospective, Non-Interventional Study of Dabraf- 2 enib Plus Trametinib in Unselected Patients with Unresectable 3 or Metastatic BRAF V600-mutant Melanoma" is very well designed and methods are explained properly.
English writing is pretty well. Results and discussion sections are very well written.
Diagram quality can be improved.
Author Response
Please find enclosed below (inside the Word document) our replies

Round 2
Reviewer 1 Report
Please indicate clearly the changes you made for the next submissions (number of line/highlight)